

# Response of Trace Gases to the Disrupted 2015-2016 Quasi-Biennial Oscillation

Olga V. Tweedy[1], Natalya A. Kramarova[2,4], Susan E. Strahan[3,4], Paul A. Newman[4], Lawrence Coy[2], William J. Randel[5], Mijeong Park[5], Darryn W. Waugh[1], and Stacey M. Frith[2,4]

[1]Johns Hopkins University, Baltimore, MD, USA
[2]Science Systems and Applications Inc.,Lanham, MD, USA
[3]Universities Space Research Association, Greenbelt, MD, USA
[4]NASA GSFC, Greenbelt, MD, USA
[5]National Center for Atmospheric Research, Boulder, CO, USA

Correspondence to: Olga Tweedy (otweedy1@jhu.edu)





## Abstract

The quasi-biennial oscillation (QBO) is a quasi-periodic alternation between easterly and westerly zonal winds in the tropical stratosphere, propagating downward from the middle stratosphere to the tropopause with a period that varies from 24 to 32 months ($\sim$28 months on average). The QBO wind oscillations affect the distribution of chemical constituents, such as ozone ($O_3$), water vapor ($H_2O$), nitrous oxide ($N_2O$) and hydrochloric acid (HCl), through the QBO induced meridional circulation. In the 2015-2016 winter, radiosonde observations revealed an anomaly in the downward propagation of the westerly phase, which was disrupted by the upward displacement of the westerly phase from $\sim$30 hPa up to 15 hPa, and the sudden appearance of easterlies at 40 hPa. Such a disruption is unprecedented in the observational record from 1953-present. In this study we show the response of trace gases to this QBO disruption using $O_3$, HCl, $H_2O$ and temperature from the Aura Microwave Limb Sounder (MLS) and total ozone measurements from the Solar Backscatter Ultraviolet (SBUV) Merged Ozone Data Set (MOD). Results reveal development of positive anomalies in stratospheric equatorial $O_3$ and HCl over $\sim$ 50-30 hPa in May-September of 2016 and a substantial decrease in $O_3$ in the subtropics of both hemispheres. The SBUV observations show near record low levels of column ozone in the subtropics in 2016, resulting in an increase of surface UV index during northern summer. Furthermore, cold temperature anomalies near the tropical tropopause result in a global decrease in stratospheric water vapor.

## 1 Introduction

The quasi-biennial oscillation (QBO) is a quasi-periodic alternation between easterly and westerly zonal winds in the tropical stratosphere that is driven by a broad spectrum of vertically propagating Kelvin and mixed Rossby-gravity waves along with smaller-scale gravity waves (Lindzen and Holton, 1968; Holton and Lindzen, 1972; Dunkerton, 1997). The alternating wind regimes (i.e., the easterly and westerly phases) propagate downward from the



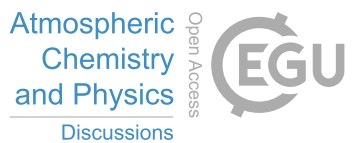

middle stratosphere to the tropopause with a period that varies from 24 to 32 months (~28 months on average).

There is also a QBO in ozone (O$_3$), which was first observed by Funk and Garnham (1962) in Australian mid-latitude total column O$_3$ observations. Ramanathan (1963) showed the connection between total O$_3$ and the QBO in a series of ground stations spanning both hemispheres, but most importantly noted the total ozone out-of-phase relationship between mid-latitudes and the equator. Angell and Korshover (1964) found a QBO signal in Shanghai (31.2°N) total O$_3$ observations in the 1932-1942 period. Zawodny and McCormick (1991) used satellite O$_3$ profile observations to show the ozone QBO vertical structure from 20-50 km and 50°S-50°N. Randel and Wu (1996) used numerical techniques to filter the QBO ozone structure showing the equatorial and mid-latitude "out-of-phase" relationship, but also revealing the seasonal synchronization between the equatorial QBO and the large amplitude winter-to-spring QBO anomalies in both hemispheres.

Because most O$_3$ is found in the lower stratosphere where its lifetime is more than 1 year, the tropical O$_3$ distribution is strongly controlled by the tropical lower stratosphere transport (Ling and London, 1986). Gray and Pyle (1989) used a two-dimensional (latitude vs. altitude) model to simulate the relationship between winds, temperatures, and the O$_3$ distributions. Those modeled relationships were confirmed by the observations of Zawodny and McCormick (1991). The Gray and Pyle (1989) simulation revealed that the wave-induced QBO drove a secondary meridional circulation which modulated the O$_3$ distribution. Assimilated meteorological data and modern transport models confirm these early results, and satellite instruments such as the Microwave Limb Sounder (MLS) have shown additional QBO impacts on water (H$_2$O), hydrochloric acid (HCl), nitrous oxide (N$_2$O), and carbon monoxide (CO) (Schoeberl et al., 2008).

The QBO meridional circulation develops between the tropics and subtropics (from the equator to ~30°N and 30°S) to maintain the thermal wind balance between the descending QBO wind shear and its temperature anomaly. At the equator, westerly shear (westerlies aloft and easterlies below) is in balance with a downward, adiabatically warmed perturbation, while easterly shear (easterlies aloft and westerlies below) produces an upward, adi-



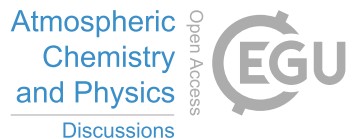

abatically cooled perturbation. The enhanced upwelling during easterly shear and reduced upwelling during westerly shear in the tropics are mass balanced by the changes in the subtropical descent. The circulation is 'completed' by the equatorial divergence/convergence of air at the levels of maximum easterly/westerly winds (Choi et al., 2002). The QBO-induced meridional circulation acts on local trace gas gradients to modify their distributions (Gray and Chipperfield, 1990). $O_3$ responds with increased/decreased values in the tropics and decreased/increased values in the extratropics during descending westerly/easterly shear.

QBO effects on composition are found throughout the extratropics. QBO-driven column $O_3$ anomalies originating in the southern subtropics in early winter reach 60°S by the early spring (Gray and Ruth, 1993; Randel and Wu, 1996; Kinnersley and Tung, 1998). Strahan et al. (2015) also showed a transport pathway by which the midlatitude middle stratosphere QBO signal affects polar $O_3$ depletion by modulating Antarctic inorganic chlorine.

The QBO has been widely analyzed because it is a major source of stratospheric $O_3$ inter-annual variability (Baldwin et al., 2001), and the QBO in total column $O_3$ is a dominant factor controlling inter-annual variations of surface ultraviolet levels (Udelhofen et al., 1999). Further, the detection and attribution of long-term $O_3$ changes caused by ozone depleting substances (ODSs) requires accurate statistical models that include QBO regression terms in order to remove the QBO driven natural $O_3$ variability and thereby reveal the residual ODS forced ozone depletion (e.g. Stolarski et al., 1991). Hence, investigating the QBO driven variability is fundamental to understanding $O_3$ levels and trends, and the resulting changes to surface ultraviolet (UV) radiation.

During the Northern Hemisphere (NH) winter of 2015-2016, radiosonde observations revealed that the normal downward propagation of the QBO westerly phase was disrupted by upward propagation of westerlies from ∼30 hPa up to 15 hPa, and the sudden appearance of easterlies at 40 hPa (Newman et al., 2016; Osprey et al., 2016). This disruption began in November 2015, and the easterlies were fully developed by March 2016. Such a disruption is unprecedented in the equatorial wind observational record from 1953-present. Osprey et al. (2016) showed that this anomalous event was linked to the transport of easterly momentum from the northern extratropics into the equatorial region, and Coy et al. (2017),





using meteorological analysis fields beginning in 1980, showed that the 2015-2016 tropical easterly momentum flux had the largest values over the December-February. None of these studies examined the changes in $O_3$ or other trace gases during 2015-2016.

In this study we investigate the response of stratospheric trace gases to the 2015-2016 event. We quantify the impact of the disruption on $O_3$ and other trace gases and further compare their observed changes to the expected behavior due to the QBO in the absence of the disruption. Furthermore, we examine the interannual variations in total ozone and water vapor.

## 2 Methods and Data

We use Aura MLS version 4.2 Level 2 measurements of temperature (T), $O_3$, and HCl from January 2005 to September 2016 between 10-100 hPa. $O_3$ and T are reported on a vertical resolution grid with 12 pressures per decade; HCl is reported on 6 pressures per decade. $O_3$ and HCl have vertical resolution of $\sim 3$ km in the pressure range used in this analysis. $O_3$ accuracy varies from 50 to 300 ppb in this range while HCl accuracy is estimated at $\sim$10%. The vertical resolution of T is $\sim$4 km; MLS temperatures have a -1 K bias in the stratosphere with respect to correlative measurements. Details on MLS measurements, data quality and improvements over previous versions can be found in MLS v4.2 data quality document (Livesey et al., 2015).

We examine total column $O_3$ during the anomalous QBO event using total $O_3$ observations from the Solar Backscatter Ultraviolet (SBUV) Merged Ozone Data Set (MOD). The MOD is constructed from monthly zonal mean ozone profiles by individual SBUV instruments, providing the longest available satellite-based time series of profile and total $O_3$ from a single instrument type (Frith et al., 2014). The MOD used here includes observations from January 1980 to the present to evaluate the temporal and spatial distribution of total $O_3$.

We use monthly averaged analyses of meteorological data on constant pressure levels from the Modern-Era Retrospective analysis for Research and Applications- Version 2,



(MERRA-2) (Bosilovich et al., 2015) to determine the vertical wind shear and QBO phase. The MERRA-2 analysis begins in January 1980. Coy et al. (2016) showed that MERRA-2 produces a realistic QBO from 1980-2016, a period encompassing 15 QBO cycles. We show changes in the meridional circulation due to the disrupted QBO using the vertical component of the MERRA-2 residual mean meridional circulation ($\overline{w}*$), which is calculated using 3-hourly output.

To determine the impact of the 2015-2016 disruption on the distribution of trace gases, we create a QBO composite ('QBO climatology') for each analyzed dynamical variable (T, zonal winds ($\overline{u}$), and upwelling by the residual circulation ($\overline{w}*$)) and trace species ($O_3$, HCl, and total $O_3$). These QBO composites include all available data except for 2015-2016. We composite based on the month of change from zonal mean easterly (negative) to westerly (positive) vertical wind shear at 40 hPa. This is identified by month '0' in the figures. Compositing based on this criterion emphasizes that chemical trace gases are most closely related to the changes in the wind shear ($\frac{\partial \overline{u}}{\partial z}$ or $\overline{u}_z$), not the zonal wind ($\overline{u}$) (Baldwin et al., 2001). Compositing dates (month 0) for each QBO cycle are listed in Table 1. Prior to compositing, we use monthly $\overline{u}$ data to compute $\overline{u}_z$ as a first vertical derivative of an unevenly-spaced array of $\overline{u}$ using three-point (quadratic) Lagrangian interpolation. The annual cycle was removed from T, $\overline{w}*$ and trace gases to better isolate QBO variations, and values are shown in percent difference relative to the monthly climatology, except for the total column $O_3$, which is shown as absolute difference in Dobson Units (DU). The difference between '2015-2016' and the composite highlights anomalies due to the 2015-2016 event.

Empirical orthogonal function (EOF) analysis has been applied to specify the instantaneous state of the QBO (Wallace et al., 1993). The two leading EOFs (EOF1 and EOF2) were derived from the deseasonalized monthly mean zonal mean wind data from Singapore radiosondes.

Interannual changes in global stratospheric water vapor and cold point tropical tropopause temperatures were analyzed by forming timeseries from multiple observational data sets. Data were deseasonalized to isolate anomalies due to the QBO. Time series of stratospheric water vapor anomalies were derived from combined HALOE (1991-2005) and Aura

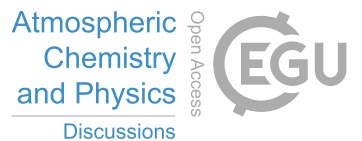

MLS (2004- 2016) satellite measurements. These data represent near-global (∼60°N-S) averages in the lower stratosphere (83 hPa). HALOE and MLS data were combined using the overlap period during 2004-2005. Cold point temperatures are derived from radiosonde data (1991-2016) and GPS radio occultation data (2001–2016). More details of the data and analysis are provided by Randel and Jensen (2013).

## 3  Results

### 3.1  The response of the equatorial stratosphere to the anomalous QBO event

Prior to 2015, wind observations show the robust features of the QBO's zonal wind pattern of descent in the middle and lower equatorial stratosphere (Newman et al., 2016; Coy et al., 2017). The $\overline{u}$ composite in Fig. 1 (top, column a) shows typical descending easterlies (blue) and westerlies (red) with the zero wind shear (thick solid black contours). The alternating regime of downward propagating wind shear leads to a modification of lower stratospheric tracers. Composites of tropical $O_3$ and HCl (Fig. 1, column a) show decreases/increases in mixing ratios, relative to the climatological seasonal values, during negative (easterly)/positive (westerly) wind shear. This $O_3$ and HCl behavior results from the QBO-induced ("secondary") meridional circulation, acting on local gradients of these chemical tracers (Gray and Chipperfield, 1990). A downward (adiabatically warmed) perturbation (decreased $\overline{w}*$ and increased T) is associated with descending westerly wind shear (positive $\overline{u}_z$) while an upward (adiabatically cooled) perturbation is associated with easterly shear (negative $\overline{u}_z$). Since the tracer's tendency ($\overline{\chi}_t$) is proportional to $-\overline{w}*\overline{\chi}_z$ and mixing ratios of both chemical species increase with height in the lower stratosphere (positive vertical gradient, $\overline{\chi}_z$), $O_3$ and HCl decrease with time when $\overline{w}*$ increases. The opposite is true for a downward perturbation (decreased $\overline{w}*$). This is supported by the good agreement between the analyzed $\overline{w}*$ and observed T, $O_3$ and HCl composites with vertical wind shear. As discussed in the introduction, horizontal transport completes the circulation and also contributes to some changes in tropical stratospheric composition (not shown).





In late 2015 westerlies were displaced upward between 30 and 15 hPa and anomalous easterlies developed at ∼40 hPa in early 2016 (Newman et al., 2016; Coy et al., 2017), see Fig. 1b (top). During the northern spring the anomalous ascending westerlies reverted back to a more typical descent, reaching 50 hPa in September 2016. The vertical residual velocity, $\overline{w}*$ (Fig. 1b bottom), responded to the changes in equatorial zonal winds during 2015-2016 with decreased upwelling in association with the westerly shear and increased upwelling below the easterly maximum. A strong positive temperature perturbation developed in the 50-30 hPa layer (westerly shear zone) due to this reduced upwelling, while a strong negative perturbation developed in the easterly shear zone – due to the enhanced upward motion. Although analyzed $\overline{w}*$ is noisy and involves greater uncertainty because of its highly derived nature, the excellent agreement between the wind shear and temperature changes (Fig. 1b bottom panels) provide evidence of secondary circulation changes resulting from the anomaly.

The circulation anomalies created by the 2015-2016 event altered stratospheric composition patterns (Fig. 1b, $O_3$ and HCl) relative to the composites seen in the left column. Changes in $O_3$ and HCl are in good agreement with changes in the wind shear and temperature (and thus $\overline{w}*$). Prior to November 2015, both trace gases were following similar tendencies to their composites. Beginning February 2016 (black vertical line in Fig. 1b), $O_3$ and HCl increased between 50-30 hPa (due to the reduced upward motion) and decreased below 50 hPa (due to the enhanced upward motion). When the composites are subtracted from the last QBO cycle (Fig. 1c), QBO-induced anomalies in T, $O_3$ and HCl are seen to be co-located with the changes in wind shear during the last QBO cycle, indicating that their changes were driven by the 2015-2016 QBO disruption.

The unprecedented nature of the 2015-2016 event is demonstrated in Fig. 2 showing "the orbits of the QBO" in two dimensional phase space, based on the projections on two leading empirical orthogonal functions (EOF1 and EOF2), following Wallace et al. (1993). QBO orbits are used to quantify the amplitude and phase propagation among the QBO cycles. Each point in this figure describes the instantaneous state of the QBO, described by the amplitude and phase angle of the vector in polar coordinates and specified in terms of vari-



ables (EOF1 and EOF2) that define the vertical structure of the zonal wind. EOF1 reflects the negative correlation between zonal wind fluctuations at 10 and 70 hPa while EOF2 indicates the variability at intermediate levels. Time progression corresponds to counter-clockwise transits and each orbit corresponds to an individual QBO cycle.

Prior to 2016, the QBO orbits (blue dots in Fig. 2) are roughly circular and data points are distributed uniformly along the orbits, indicating the remarkably uniform structure and nearly constant amplitude of the QBO in this record (1987-2016). Based on this stability, EOF1 and EOF2 are commonly used to isolate the variability related to the QBO when deriving ODS-induced changes in long-term ozone records. The repetitive QBO pattern was disrupted in 2016, as shown by the red points that deviate from the regular circular pattern. Such disruptions add unpredictable variability to the time series, directly interfering with the ability to determine regression-based trends in stratospheric ozone.

## 3.2 Latitudinal changes in ozone

The stratospheric impact of the 2015-2016 event extends into the extratropics. Figure 3 shows the evolution of $O_3$ for the composite (left), 2015-2016 (middle) and their difference (right). The top panels of Fig. 3 (a, b, and c) show MLS $O_3$ at 38 hPa, the pressure level of maximum $O_3$ anomaly during the NH summer of 2016. As shown in the composite (Fig. 3a), the positive $O_3$ perturbation during the westerly shear in the tropics and the negative perturbations in the extratropics are replaced by anomalies of opposite signs as the wind shear reverses to easterly 12-14 months later. However, in 2016 (Fig. 3b) the 40 hPa westerly shear changes to a weak easterly shear for only a short time interval (Jan-Apr) before switching back to westerly shear (note that the wind shear at 30 hPa remains westerly). $O_3$ responds by decreasing/increasing in mixing ratios during the easterly/westerly shear changes. The $O_3$ anomalies due to the 2015-2016 event are highlighted in Fig. 3c, which show large differences after February 2016 (black line). A strong negative tropical $O_3$ perturbation developed by early spring 2016, propagating to the extratropics in both hemispheres by the end of the NH summer. In the equatorial region, a positive perturbation replaced the negative $O_3$ perturbation as the wind shear switched back to westerly.

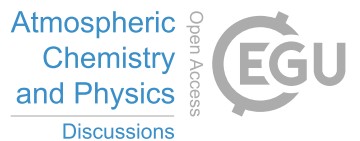

The 2016 NH summer positive tropical $O_3$ anomaly at 30-50 hPa - the level of maximum $O_3$ number density - contributed to substantial changes in the total column $O_3$. As in the 38 hPa $O_3$ from MLS (Fig. 3, top), the typical QBO behavior of SBUV total $O_3$ (Fig. 3d) contrasts to the anomalous 2015-2016 behavior (Fig. 3e), with their difference (Fig. 3f) highlighting the 2016 anomalies. The SBUV total $O_3$ QBO composite is based on 14 transitions of wind shear at 40 hPa (excluding 2015-2016). Total $O_3$ was deseasonalized and values are shown as absolute difference from the monthly climatology (in Dobson Units). This SBUV composite captures the major features of the typical QBO and the $O_3$ perturbations.

Prior to February 2016, Fig. 3f shows small total $O_3$ differences except for the large midlatitude anomalies from 0 to +6 months in the southern (negative anomaly) and northern (positive anomaly) hemispheres. However, only the negative anomaly in the southern midlatitudes is significant at the 2-sigma level(not shown). The cause of these anomalies prior to the 2015-2016 disruption remains unclear and is a subject of ongoing investigation. After February 2016 (black line in Fig. 3e), there is a large decrease in total ozone in the extratropics and midlatitudes of both hemispheres and the total $O_3$ differences between composite and last QBO cycle (Fig. 3f) are very similar to the differences in 38 hPa $O_3$ from MLS (Fig. 3c). This strongly suggests that the 2015-2016 event had a significant impact on both tropical and extratropical total $O_3$.

### 3.3 Temporal and spatial morphology of ozone in April and August 2016

Large negative $O_3$ anomalies in the lower stratosphere start in April 2016 in the tropics and propagate to the extratropics by August 2016. Note, these two months occur at 11 and 15 months after month 0 on the "composited" time axes and are indicated by arrows in Fig. 3. Therefore, we compare the latitudinal and vertical extent of the QBO-induced anomalies in T and $O_3$ during these two months (Fig. 4b and Fig. 4d) to the expected behavior based on the composite eleven (+11 mon) and fifteen (+15 mon) months after the wind shear reversal (Fig. 4a and Fig. 4c). In agreement with Fig. 1b, in April 2016 the anomalous easterly shear below 40 hPa (dashed horizontal line in Fig. 1b) leads to strong negative T and $O_3$ perturbations in the tropics while the appearance of ascending westerly shear leads to



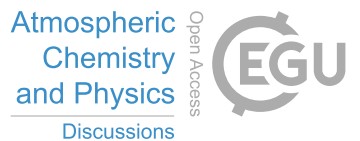

weak tropical T and $O_3$ increases between 20-40 hPa. By August 2016, the westerly shear strengthens and descends to 30-50 hPa resulting in strong positive T and $O_3$ perturbations in this layer, while the easterly shear below 50 hPa leads to negative perturbations in the equatorial (10°N-10°S) stratosphere. This is consistent with our understanding of trace gas response to changes in tropical upwelling. The consistency of $O_3$ and T anomalies during 2015-2016 is evidence of circulation changes.

In the subtropics, the deseasonalized $O_3$ QBO signature is out-of-phase with that at the equator, in agreement with Fig. 3b. In August 2016, strong negative $O_3$ perturbations develop during the NH summer on both sides of the equator (although much stronger in the winter southern hemisphere) as a response to the QBO-induced meridional circulation. In the composites at +11 and +15 months (Fig. 4a and Fig. 4c) anomalies in MLS $O_3$ and T are opposite to those observed in April and August 2016 (Fig. 4b and Fig. 4d) due to the descending easterly shear. Thus, $O_3$ is responding as expected to a QBO-induced meridional circulation but this period is anomalous with respect to a normal QBO progression.

The observed 2015-2016 anomalies are unique in the total $O_3$ record. Figure 5 compares the latitudinal distribution of deseasonalized total $O_3$ from April and August 2016 (in red) to the total $O_3$ composite (the average from the 14 composited QBO cycles) shown in Fig. 3d at +11 and +15 months respectively (in black). Total $O_3$ from each individual QBO cycle included in the composite is shown in blue, with light blue shading indicating the range of total $O_3$ from all QBO cycles (excluding 2015–2016). In the absence of the disruption, we expect total $O_3$ at +11 (Fig. 5a) and +15 (Fig. 5b) months to lie within the blue shaded range of past observations. Instead, in April 2016 total $O_3$ is lower than during other QBO cycles in the NH tropics (10°N - 20°N). Furthermore, during August 2016 total ozone is higher at the equator and lower/near the edge in the extratropics between 10°S-40°S / 30°N-50°N compared to other QBO cycles.

Calculations suggest that anomalously low total column $O_3$ at 22.5°S in August 2016 increased the monthly zonal mean surface clear-sky UV index by ~8.5 % compared to the 36-yr mean (Newman and McKenzie, 2011). Increased surface UV radiation has a harmful effect on health by damaging cells, DNA, and increasing the risk of developing skin cancer.





Increased exposure to UV in plants leads to enhanced plant fragility, growth limitation, and yield reduction affecting our ability to secure food production (Caldwell et al., 1995; Tevini, 1993).

### 3.4 QBO-driven changes in total ozone and water vapor in the context of long-term time series

Examination of the interannual variations in SBUV monthly and zonal mean total $O_3$ shows very low total $O_3$ values in the extratropics during spring and summer of 2016 compared to other years within this observational record. Figure 6 displays the time series for April (top panels) and August (bottom panels) total $O_3$ values in the northern and southern extratropics. The regions shown are locations with large anomalies in Fig. 3f. The individual 1-sigma error estimates are shown as the vertical gray bars in Fig. 6, while the horizontal line is the 2016 value. Each of these plots show that the 2016 value was the record or near record low in the more than 40 years of the SBUV data.

Near record low total $O_3$ in the extratropics during the spring-summer 2016 is due to the 2015-2016 QBO disruption event. As shown in Fig. 3e, beginning about February 2016 the disruption in the descent of easterly zonal winds led to lower $O_3$ values in both the northern and southern extratropics and persisted into the fall of 2016. The ozone anomalies, shown in Fig.3e, are up to -12 DU at 12.5°N in April 2016, and -15 DU at 17.5°S in August 2016. This strongly contrasts with the expected behavior (Fig. 3d) that would have been either near zero or small positive anomalies.

The 2015-2016 QBO event also significantly impacted the global amount of stratospheric water vapor in 2016. $H_2O$ enters the stratosphere from the troposphere primarily in the tropics. The amount of $H_2O$ in the stratosphere is controlled by the tropical tropopause temperature ("cold point tropopause temperatures", or $T_{cp}$) with colder $T_{cp}$ resulting in less $H_2O$ entering the tropical stratosphere from the troposphere. Figure 7 demonstrates very strong correlation of stratospheric water vapor anomalies with $T_{cp}$ (also see Randel and Jensen, 2013). In 2016, cold tropical tropopause temperatures (in balance with anomalous easterlies) led to a global decrease in the stratospheric $H_2O$ in October-December 2016.

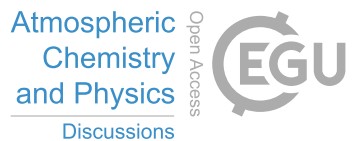

Global $H_2O$ in November is amongst the lowest in the record (1992-2016) due to very low $T_{cp}$.

## 4  Concluding remarks

This study demonstrates that the 2015-2016 QBO disruption had a substantial impact on the composition of the stratosphere. It led to a modified circulation that reduced the equatorial upward circulation in association with the positive (westerly) shear, while the negative shear below the easterly maximum led to enhanced upward motion. Following the appearance of the disruption in February 2016, there were two layers of zonal wind shear in the tropics. Westerly shear in the 30-50 hPa layer was linked to increased temperature and decreased upwelling, resulting in positive perturbations in $O_3$ and HCl. The easterly shear from the disruption in the 50-100 hPa layer produced negative temperature perturbations in association with increased tropical upwelling, inducing negative perturbations in $O_3$ and HCl. Cold temperature anomalies extended to the tropopause level in late 2016, resulting in decreases in global stratospheric water vapor. Because the ozone number density maximum is in the 50-30 hPa layer, the QBO disruption increased total $O_3$ at the equator.

The decrease in tropical ascent during the disruption was balanced by reduced downwelling in the extratropics. This reduced extratropical downward motion decreased $O_3$ in those regions (although the horizontal component to this circulation contributes as well). In this study we focused mostly on $O_3$ changes, however, the response of other long-lived tracers such as HCl and $N_2O$ is consistent with the QBO meridional circulation induced by the disrupted QBO. While HCl anomalies are consistent with the $O_3$ anomalies, the $N_2O$ anomalies have an opposite sign due to the negative vertical gradient of this tracer. The similarities in the responses of temperature and observed changes in chemical trace gases to the QBO disruption shows that these composition changes are primarily dynamically driven. Trace gases show perturbed behavior compared to the past, but their response is consistent with our understanding of the QBO-induced meridional circulation.


It is unclear if this QBO disruption is an event of great rarity or if similar events will re-occur. Similar disruptions with the same timing could potentially alter ozone and trace gas distributions, affecting the stratospheric climate and making it more difficult to accurately estimate climate trends. For example, a series of disruptions could drive a downward ozone trend and lead to a long-term increase in the surface UV index during the peak of northern summer.

At present, numerical models are unable to predict such events (Osprey et al., 2016), pointing to incomplete understanding of QBO forcing mechanisms. The model failures could result from missing processes, poor representation of necessary wave forcings, or resolution. Osprey et al. (2016) pointed out that only one event similar to the observed during 2016 was identified among the available models that produce an internally-generated QBO. Our inability to simulate and/or predict a disrupted QBO will add uncertainty to future predictions of ozone and other chemical constituents from coupled chemistry climate models, as well as limit our ability to resolve statistically-significant ODS-related changes in the observed $O_3$ record. This event, whether unique or the first of many QBO disruptions, emphasizes the crucial need to continue collecting and evaluating high quality satellite measurements to trace the impact of stratospheric dynamical changes.

*Acknowledgements.* Authors acknowledge, and thank, Luke Oman for providing us with vertical component of the residual circulation, $\overline{w}*$, calculated from MERRA-2 reanalysis product and all those who are members of the MLS and SBUV data science and support teams. Olga Tweedy is supported by the National Science Foundation Graduate Research Fellowship Program under Grant No.DGE-1232825. Support for this study was also provided under the NASA Atmospheric Composition Modeling and Analysis Program. NCAR is operated by the University Corporation for Atmospheric Research, under sponsorship of the National Science Foundation.

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





**Fig. 1.** The rows show the MERRA-2 zonal mean zonal wind component, $\overline{u}$ (m s$^{-1}$), deseasonalized MLS O$_3$, HCl, temperature (T) and vertical component of the MERRA-2 residual circulation ($\overline{w}*$), as a function of time and pressure (in percent change from long term monthly averages), averaged over 5°S–5°N. The left column (a) shows the composite of the easterly-to-westerly shear transitions based on 4 shear transitions at 40 hPa. The middle column (b) shows the 2015-2016 QBO cycle, which includes the data from April 2014 to September 2016, with month 0 in May 2016. The right column (c) shows the difference between the 2015-2016 event and the climatology (b-a). The thick black contours denote the zero wind shear. The horizontal dashed line indicates the 40 hPa level while vertical line indicates February 2016.

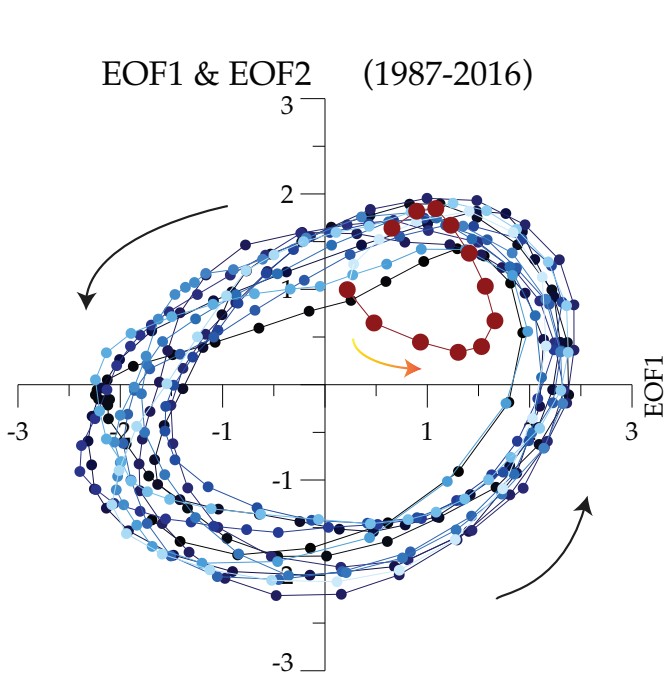

**Fig. 2.** Phase space diagram of the projection of the monthly equatorial zonal wind anomalies onto spacial structures EOF1 and EOF2. Time progression coincides with counterclockwise orbit transits. Dots represent each month from 1987-2016. Different shades of blue indicate different years from 1987-2015 (from darker to lighter) while red dots correspond to 2016.

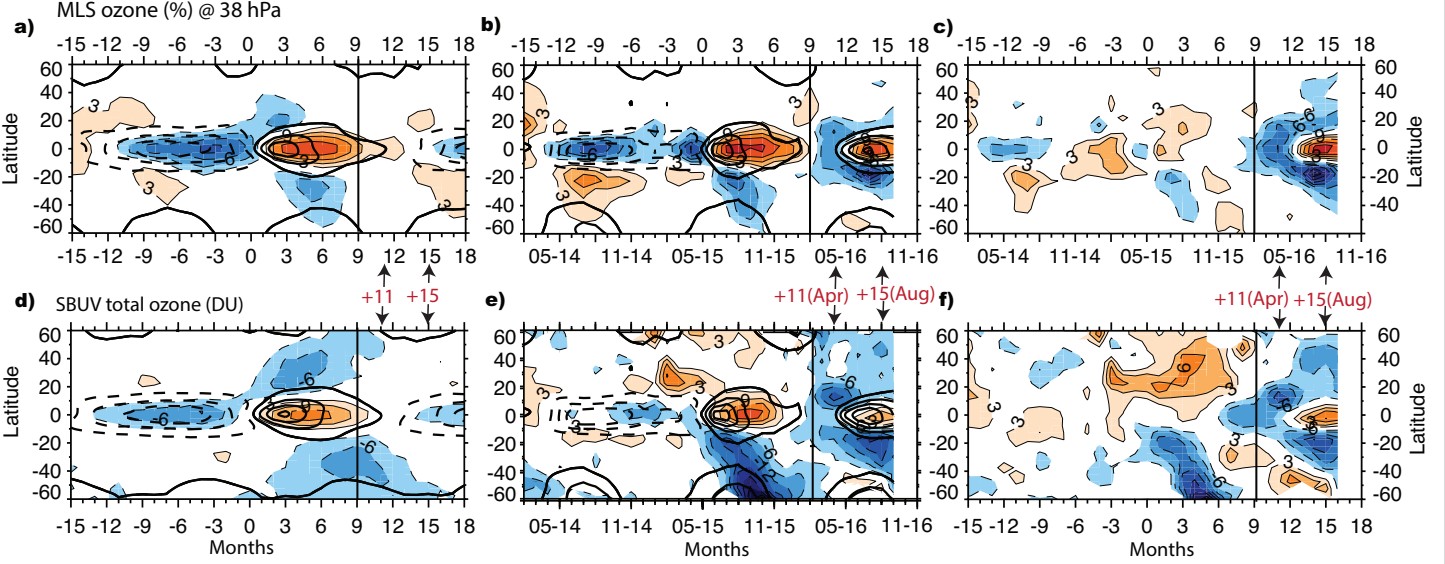

**Fig. 3.** Latitude and time evolution of MLS ozone at 38 hPa (top row) for (a) the composite, (b) 2015-2016, and (c) their difference (b-a), highlighting the anomalies due to the disruption. MLS ozone values are shown in percent change from long term monthly averages with contour intervals every 3 % (zero contour is omitted). The bottom row shows the deseasonalized SBUV total ozone (in Dobson Units, contour intervals every 3 DU) for d) the composite, e) 2015-2016, and f) their difference (e-d). Black thick solid and dashed contours show westerly and easterly vertical wind shear respectively for (a and d) the composites and (b and e) 2015-2016. MLS (SBUV) composites are based on 4(14) transitions from easterly to westerly vertical wind shear at 40 hPa. Vertical black line highlights +9 months after windshear reversal from negative to positive (month 0), corresponding to February 2016 in (b) and (e) while arrows indicate ozone at +11 and +15 months after month 0, corresponding to April 2016 and August 2016 in (b) and (e).



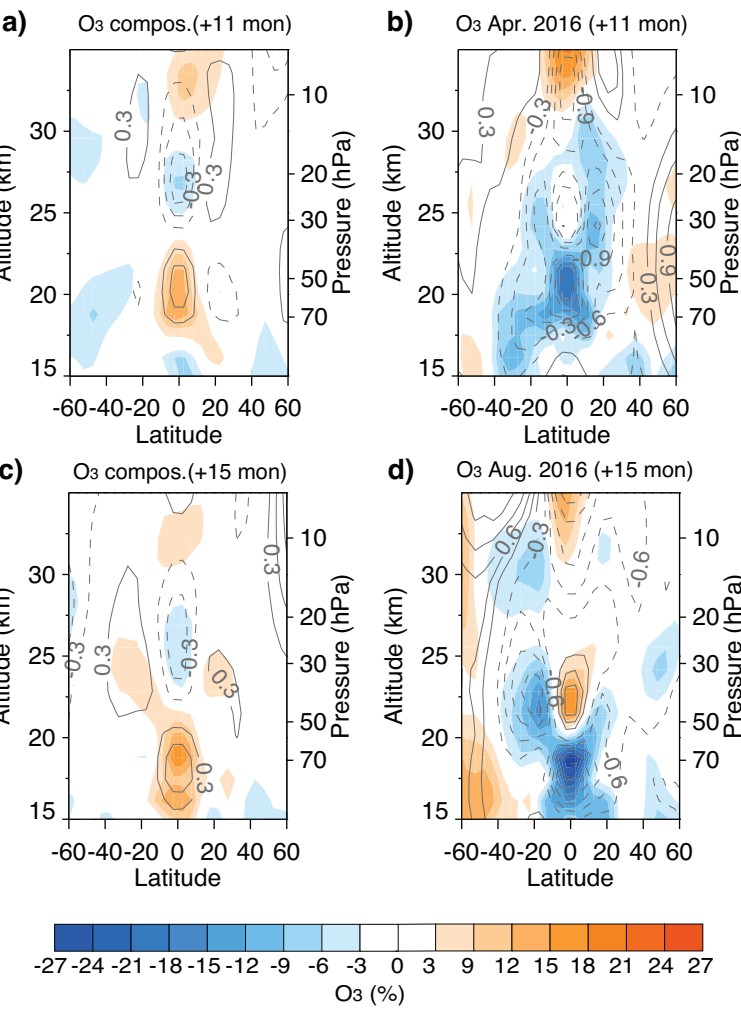

**Fig. 4.** Latitude-height cross sections of deseasonalized MLS ozone (filled) and temperature (gray contours) in the composite (a) 11 and (c) 15 months after the wind shear reversal based on 4 QBO cycles and during (b) April (+11 mon.) and (d) August 2016 (+15 mon.). Ozone and temperature values are shown in percent change from long term monthly averages with contour intervals every 3% and 0.3% respectively (zero contour is omitted).

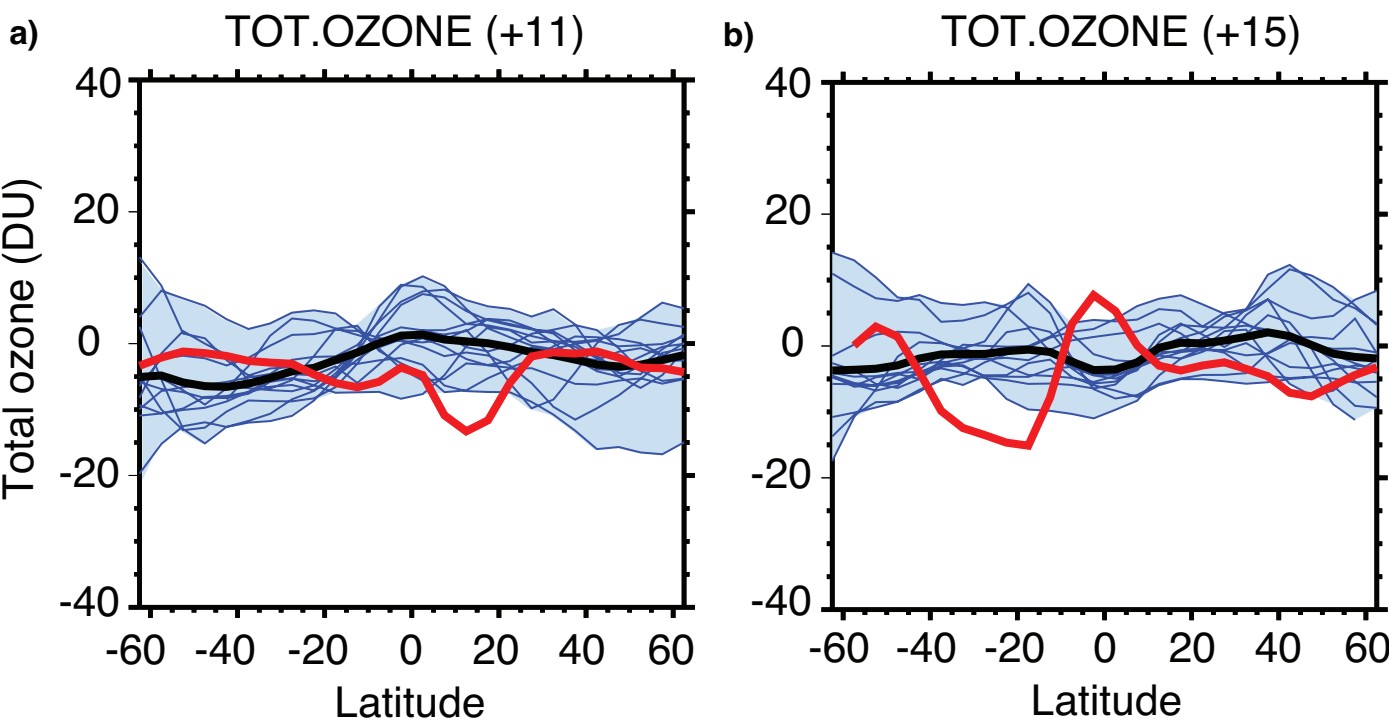

**Fig. 5.** (a) Deseasonalized SBUV total ozone (in Dobson Units) as a function of latitude 11 months after wind shear reversal from easterly to westerly from fourteen QBO cycles prior to 2015-2016 (blue lines), the composite (black line) based on 14 QBO cycles and 2015-2016 (April 2016, red line). (b) The same as in (a) only for total ozone at +15 months, corresponding to August 2016. The blue shading shows the observed $O_3$ range at +11 and +15 months respectively for all 14 QBO cycles (excluding the 2015–2016 event).





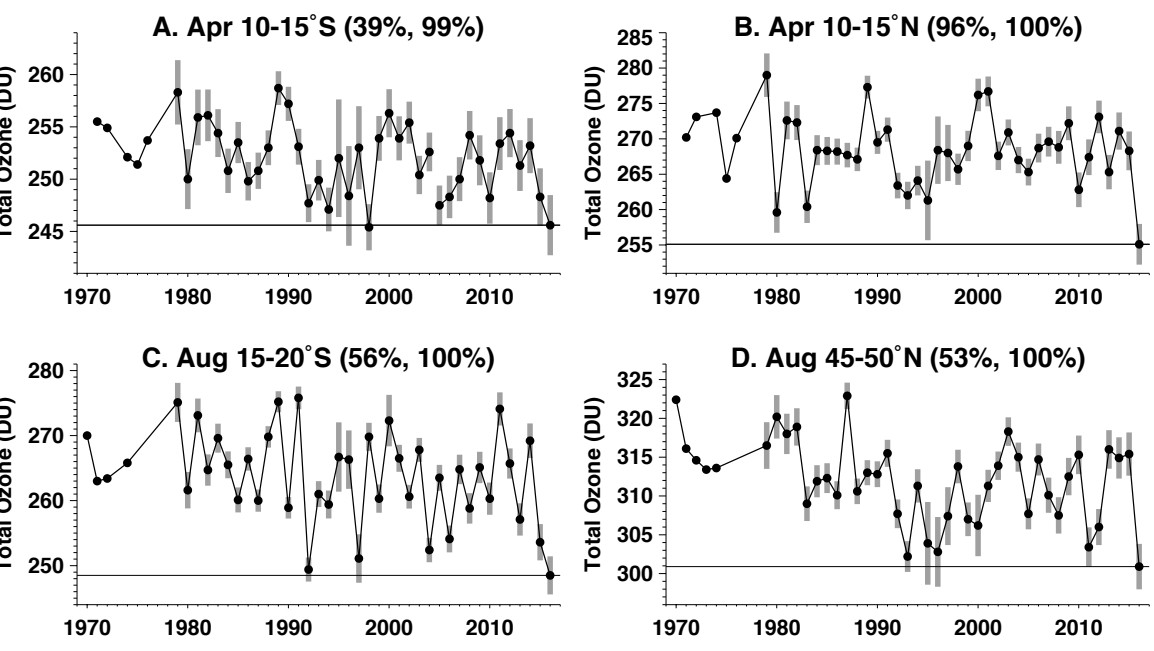

**Fig. 6.** SBUV total ozone (in Dobson Units) timeseries from 1970-2016 for April, averaged over (a) 10°S -15°S and (b) 10°N -15°N, and for August, averaged over (c) 15°S -20°S and (d) 45°N -50°N . Vertical bars show 1-sigma uncertainties in the measurements. The horizontal line shows the total ozone value in April or August 2016 and the panel titles show the percentage estimates of the 2016 value being the lowest and amongst the lowest 20% of all values. The probability that the 2016 values were record lows was estimated using a 10,000 Monte Carlo simulations of the monthly means in the time series (Frith et al, 2014).





**Fig. 7.** Observed variations in lower stratospheric water vapor and tropical cold-point tropopause temperatures from satellite measurements over the period 1992–2016. Water vapor data are deseasonalized near-global averages at 83 hPa from combined HALOE and MLS satellite measurements. Each dot represents a monthly average. Temperatures are deseasonalized anomalies derived from radiosonde data (black line) and GPS radio occultation data (red line, for 2001–2016). Vertical bars are one sigma standard deviations of the monthly averages.





**Table 1.** the QBO composite dates (month 0)

| QBO cycle | 1 | 2 | 3 | 4 | 5 | 6 | 7 | 8 |
|---|---|---|---|---|---|---|---|---|
| Month 0 | 1982-06 | 1984-10 | 1987-06 | 1989-12 | 1992-06 | 1994-09 | 1996-11 | 1998-09 |
| QBO cycle | 9 | 10 | 11 | 12 | 13 | 14 | 15 | |
| Month 0 | 2001-10 | 2003-12 | 2006-02 | 2008-01 | 2010-06 | 2012-12 | 2015-05 | |