# Peer review of "Response of Trace Gases to the Disrupted 2015-2016 Quasi-Biennial Oscillation"

_Atmospheric Chemistry and Physics, 2017_

## Referee Comment (RC1) · Anonymous Referee #1 · 10 Mar 2017

The QBO showed a behavior in 2015-216 which has never been seen before. The development of the meteorological fields has been described elsewhere but the present paper contributes by describing how ozone and HCL changes during the event. I find that the paper contributes with new information and that it is well written. However, I have a few relatively minor points that the authors should consider before the paper is accepted.

Major comments:

In the introduction the QBO in ozone is described. However, I find this description somewhat confusing. First of all I miss a statement about if the ozone QBO is in phase with the QBO in the zonal mean wind. I am also confused about the statements about the seasonal synchronization (line 33 and 57). There is only a weak seasonal signal in

the QBO in the zonal wind.

There is only very little mention of statistical significance (line 214). The statistical significant regions should be indicated in Figs. 1 and 3 and the method to calculate the significance should be described in more details.

Minor comments:

l49: downward -> downward propagating?

l88: How can temperature and ozone have different vertical resolutions (3 and 4 km ) when they both are reported on 12 pressures per decade?

l186: The authors could be more specific here. Will the interfering make it more difficult to determine the trends? In fact, one could argue that the disruption will make it easier to establish the connection between QBO and ozone and therefore easier to determine the residual trend.

Figure 2. I am not sure this figure helps and I can not see that this analysis is used elsewhere in the paper. I would suggest that it is removed or, if the authors find it important, that also the EOFs are shown and the amount of variance they explain is mentioned. Actually, a similar figure was shown in Dunkerton 2016 (GRL 10.1002/2016GL070921) which should be cited.

[Figure]

---

## Referee Comment (RC2) · Anonymous Referee #2 · 17 Mar 2017

Response of Trace Gases to the Disrupted 2015-2016 Quasi-Biennial Oscillation

**O.V. Tweedy, N.A Kramarova, S.E. Strahan, P.A. Newman, L. Coy, W.J Randel, M. Park, D.W. Waugh, and S. Frith**

This paper examines the impact of the 2016 QBO disruption on stratospheric temperature, residual (vertical) circulation and distribution of trace gases (esp. ozone) from the equator to mid-latitudes. The paper highlights circulation and transport characteristics being dynamically consistent with the QBO anomaly. These impacts include an anomalous reduction in total ozone out to mid-latitudes (during April and August anyway) which are at near record lows. This has implications for trends in downwelling UV, if similar events were to recur more frequently in the future. The authors also highlight the possible signature of the QBO disruption in tropical cold-point tropopause

temperature and UTLS water vapor.

This is a very well written paper and does a great job of highlighting those points it considers important, without the distraction of unnecessary details. I would hope the points below can be addressed quickly as I recommend prompt publication.

**Main Points:**

I. Effect of strong polar vortex: What effect will the unusually strong polar vortex, occurring from early-mid winter 2015/2016, have on the Brewer-Dobson circulation and the redistribution of ozone? Presumably, it would create a weaker BDC and reduced downwelling outside the tropics, and so (vertical) transport of ozone at mid-latitudes. I think the conclusions of the paper also need to reflect these other environmental influences, especially as statements of attribution are being made. Here is a suitable reference for the strong vortex (and AO in general) and perhaps other conditions relevant to the 2016 QBO disruption (and redistribution of ozone):

*Cheung HHN, Zhou W, Leung MYT, Shun CM, Lee SM, Tong HW. A strong phase reversal of the Arctic Oscillation in midwinter 2015/2016: Role of the stratospheric polar vortex and tropospheric blocking. J Geophys Res Atmos. 2016; 10.1002/2016JD025288*

*Scaife AA, Comer R, Dunstone N, Fereday D, Folland C, Good E, et al. Predictability of European winter 2015/2016. Atmos Sci Lett. 2017 Feb;18(2):38–44. doi:10.1002/asl.721*

II. Effect of ENSO and subsequent interpretation of CPT and [H2O] (figure 7). The authors should acknowledge the possible influence of the 2015/2016 El Nino and the perhaps recent trends in CP temperature and pressure. One possible reference might include:

*Hu D, Tian W, Guan Z, Guo Y, Dhomse S. Longitudinal Asymmetric Trends of Tropical Cold-Point Tropopause Temperature and Their Link to Strengthened Walker Circula-*

*tion. J Clim. 2016 Nov; 29(21):7755–71. doi:10.1175/JCLI-D-15-0851.1*

III. How much of the 2016 QBO wind (during disruption) is accounted for by the first 2 EOFs (in u)? In this this regard, how meaningful is it to show PC1 and PC2 during these times?

**Minor Points:**

(line 104) One for the editorial team: superscript asterisk for TEM residual vertical velocity. Also, a reference for the TEM residual vertical velocity should be added (e.g. AHL, 1987)

(figure 2 caption) "spacial"->"spatial" deriv. spatium (latin).

(figure 7) The HALOE H20 measurements show a jump around 2001. Where does this come from? Does it affect the (statistical) significance of your results.

[Figure]

---

## Author Comment (AC1) · 3 May 2017

**Response to comments: Anonymous Referee 1**
Comments received and published: 10 March 2017

We thank the reviewer for very helpful comments. We have taken all the points raised into consideration. Our specific responses / changes are below (in *italic*).

The QBO showed a behavior in 2015-216 which has never been seen before. The development of the meteorological fields has been described elsewhere but the present paper contributes by describing how ozone and HCL changes during the event. I find that the paper contributes with new information and that it is well written. However, I have a few relatively minor points that the authors should consider before

[Figure]

the paper is accepted.

**Major comments:** In the introduction, the QBO in ozone is described. However, I find this description somewhat confusing. First of all, I miss a statement about if the ozone QBO is in phase with the QBO in the zonal mean wind. I am also confused about the statements about the seasonal synchronization (line 33 and 57). There is only a weak seasonal signal in the QBO in the zonal wind.

*We have rewritten this part of the Introduction to make it clearer. Deseasonalized ozone anomalies are out-of-phase between the tropics and extratropics; the QBO in equatorial winds are in-phase with tropical ozone anomalies but out-of-phase with extratropical ozone anomalies. "Seasonal synchronization" means the QBO influence on deseasonalized ozone anomalies in the extratropics are observed mainly in winter-spring of each respective hemisphere.*

There is only very little mention of statistical significance (line 214). The statistical significant regions should be indicated in Figs. 1 and 3 and the method to calculate the significance should be described in more details.

*The statistical significant regions are highlighted in Figure 1c and 3c,f and method to calculate the significance is described in Methods section. Since we have very limited number of QBO cases in the observational record (5 QBO cycles in MLS and 14 cycles in the SBUV total ozone, see Table 1), we don't perform any sophisticated statistical tests (degree of freedom is very small) and simply indicate regions where absolute difference between last QBO cycle and the composite larger than 2 standard deviations.*

**Minor comments:**
l49: downward -> downward propagating? - *changed*

l88: How can temperature and ozone have different vertical resolutions (3 and 4 km) when they both are reported on 12 pressures per decade?

*There is a difference between the vertical grid that data are reported and the actual vertical resolution. The vertical grid is usually finer than the actual instrumental vertical resolution. The vertical resolution is defined by the number of independent measurements (degrees of freedom for signal or DFS) that the instrument makes, and this varies between MLS temperature and ozone measurements [see MLS Version 4.2x Level 2 data quality and description document for more details]. Clarifications were made in the first paragraph of the Methods section.*

l186: The authors could be more specific here. Will the interfering make it more difficult to determine the trends? In fact, one could argue that the disruption will make it easier to establish the connection between QBO and ozone and therefore easier to determine the residual trend.

*This sentence was edited to make it more specific. Certainly, a series of similar disruptions would make it more difficult to determine the residual trends because we won't be able to rely on two EOFs to remove QBO variability in ozone timeseries. EOF 1 and 2 typically explain 96 percent of the variance while during the disruption it falls to only 71 percent. Thus, the first two EOF patterns don't match the disruption very well, with the lowest percent variance explained by the two EOFs in the entire data record occurring during the disruption.*

Figure 2. I am not sure this figure helps and I cannot see that this analysis is used elsewhere in the paper. I would suggest that it is removed or, if the authors find it important, that also the EOFs are shown and the amount of variance they explain is mentioned.

*We believe this figure is relevant and we have decided to keep it. It demonstrates how unusual and unprecedented this current QBO disruption event is. Based on Wallace et al. (1993), the first two EOFs explain 95.5 percent of the normalized variance of the deseasonalized smoothed time series of zonal winds at seven pressure levels between 70 to 10 hPa combined. We don't think additional EOF plots are necessary for this paper but discussion about the recent QBO variances is added (in Methods section)*

Actually, a similar figure was shown in Dunkerton 2016 (GRL 10.1002/2016GL070921) which should be cited.

*Our calculations follow the standard Wallace et al. (1993) EOF structures. The results look quite different from the figure in Dunkerton 2016, which came from a blog/twitter site.We are unsure exactly how the calculations were done in his case (and calculations are not clearly explained in Dunkerton 2016, on twitter, or the blog site). Our plot really doesn't suggest a 'death spiral' and plot in Dunkerton 2016 plot shows \*NO\* anomaly for the 2015-16 period! Since we don't understand this figure, how it was produced, and it was not published in peer reviewed source (except for the twitter post in Dunkerton 2016 article), we think it is best not to cite it.*

---

## Author Comment (AC2) · 3 May 2017

**Response to comments: Anonymous Referee 2**
Comments received and published: 17 March 2017

We thank the reviewer for very helpful comments. We have taken all the points raised into consideration. Our specific responses/changes are below (in *italic*).

Response of Trace Gases to the Disrupted 2015-2016 Quasi-Biennial Oscillation
O.V. Tweedy, N.A Kramarova, S.E. Strahan, P.A. Newman, L. Coy, W.J Randel, M. Park, D.W. Waugh, and S. Frith

This paper examines the impact of the 2016 QBO disruption on stratospheric temperature, residual (vertical) circulation and distribution of trace gases (esp. ozone) from the equator to mid-latitudes. The paper highlights circulation and transport characteristics being dynamically consistent with the QBO anomaly. These impacts include an anomalous reduction in total ozone out to mid-latitudes (during April and August anyway) which are at near record lows. This has implications for trends in downwelling UV, if similar events were to recur more frequently in the future. The authors also highlight the possible signature of the QBO disruption in tropical cold-point tropopause temperature and UTLS water vapor. This is a very well written paper and does a great job of highlighting those points it considers important, without the distraction of unnecessary details. I would hope the points below can be addressed quickly as I recommend prompt publication.

**Main Points:** I. Effect of strong polar vortex: What effect will the unusually strong polar vortex, occurring from early-mid winter 2015/2016, have on the Brewer-Dobson circulation and the redistribution of ozone? Presumably, it would create a weaker BDC and reduced downwelling outside the tropics, and so (vertical) transport of ozone at mid-latitudes. I think the conclusions of the paper also need to reflect these other environmental influences, especially as statements of attribution are being made. Here is a suitable reference for the strong vortex (and AO in general) and perhaps other conditions relevant to the 2016 QBO disruption (and redistribution of ozone):

Cheung HHN, Zhou W, Leung MYT, Shun CM, Lee SM, Tong HW. A strong phase reversal of the Arctic Oscillation in midwinter 2015/2016: Role of the stratospheric polar vortex and tropospheric blocking. J Geophys Res Atmos. 2016; 10.1002/2016JD025288

Scaife AA, Comer R, Dunstone N, Fereday D, Folland C, Good E, et al. Predictability of European winter 2015/2016. Atmos Sci Lett. 2017 Feb;18(2):38–

44.doi:10.1002/asl.721

*The reviewer makes a good point and we agree that a strong polar vortex occurring from early-mid winter 2015/2016 is likely to contribute (to some extent) to the observed midlatitude anomalies in late spring and through the summer. Recent study by Strahan et al. [JGR, 2016] showed that the impact of Arctic ozone depletion on the midlatitudes in spring after winters with moderate depletion (such as 2016) was about 5 DU (south of 45N). But they also found that the dynamical impact on O3 due to a strong vortex winter roughly opposed the depletion changes, resulting in very little net impact. Our analysis shows that midlatitude anomalies during boreal summer and fall of 2016 are symmetric around equator, which strongly suggests QBO induced nature of observed anomalies. We have added possible mention of the vortex playing a role in ozone anomalies in the third paragraph of section 4 ("Concluding remarks"); however, a full estimation of exact contribution is a separate study of its own.*

II. Effect of ENSO and subsequent interpretation of CPT and $H_2O$ (figure 7). The authors should acknowledge the possible influence of the 2015/2016 El Nino and the perhaps recent trends in CP temperature and pressure. One possible reference might include:

Hu D, Tian W, Guan Z, Guo Y, Dhomse S. Longitudinal Asymmetric Trends of Tropical Cold-Point Tropopause Temperature and Their Link to Strengthened Walker Circulation. J Clim. 2016 Nov; 29(21):7755–71. doi:10.1175/JCLI-D-15-0851.1

*We agree that it is possible that strong El-Nino event during 2015-2016 winter could impact CP temperature and pressure. However, the role of this ENSO event and its impact of the distribution of chemical constituents in the lower stratosphere remains an open question. As suggested by the reviewer, we have included in the text the acknowledgment of the possible influence of the 2015/2016 El Nino.*

[Figure]

III. How much of the 2016 QBO wind (during disruption) is accounted for by the first 2 EOFs (in U)? In this this regard, how meaningful is it to show PC1 and PC2 during these times?

*Figure 2 shows the smaller amplitude (closer to the center) of PC1 and PC2 during the disruption. This means the winds are either weaker than the typical QBO or not fitting the EOFs well. Looking at the variance explained by two EOFs, the answer is that it's not fitting the QBO well. Prior to the 2015, the first two EOFs explain 95.5% of the normalized variance of the deseasonalized smoothed time series of zonal winds at seven pressure levels between 70 to 10 hPa combined while during the disruption it falls to only 71%. Thus, the first two EOF patterns don't match the disruption very well, with the lowest percent variance explained by the two EOFs in the entire data record occurring during the disruption. Therefore, there shouldn't be any objection to plotting just PC1 and PC2 during the disruption as it illustrates how odd the disrupted QBO is.*

**Minor Points:**
(line 104) One for the editorial team: superscript asterisk for TEM residual vertical velocity. Also, a reference for the TEM residual vertical velocity should be added (e.g. AHL, 1987) - *changed asterisk and a reference is added*

(figure 2 caption) "spacial"->"spatial" deriv. spatium (latin). - *changed*

(figure 7) The HALOE H20 measurements show a jump around 2001. Where does this come from? Does it affect the (statistical) significance of your results.

*The drop in HALOE $H_2O$ around 2001 is a well reported phenomenon (e.g. Randel et al., JGR 2006). The cause of this drop is unclear, but previous studies have related it to changes in SSTs (e.g. Garfinkel et al., JGR, 2013) and ENSO (Brinkop et al. ACP*

*2016). This does not affect any of our conclusions.*